# Distinct Metabolic States Are Observed in Hypoglycemia Induced in Mice by Ricin Toxin or by Fasting

**DOI:** 10.3390/toxins14120815

**Published:** 2022-11-22

**Authors:** Jacob Kempa, Galen O’Shea-Stone, Corinne E. Moss, Tami Peters, Tamera K. Marcotte, Brian Tripet, Brian Eilers, Brian Bothner, Valérie Copié, Seth H. Pincus

**Affiliations:** 1Department of Chemistry and Biochemistry, Montana State University, Bozeman, MT 59717, USA; 2Animal Resources Center, Montana State University, Bozeman, MT 59717, USA

**Keywords:** hypoglycemia, liver, ricin, toxin, fasting, metabolomics, ^1^H-NMR, Kennedy pathway, multivariate analysis

## Abstract

Hypoglycemia may be induced by a variety of physiologic and pathologic stimuli and can result in life-threatening consequences if untreated. However, hypoglycemia may also play a role in the purported health benefits of intermittent fasting and caloric restriction. Previously, we demonstrated that systemic administration of ricin toxin induced fatal hypoglycemia in mice. Here, we examine the metabolic landscape of the hypoglycemic state induced in the liver of mice by two different stimuli: systemic ricin administration and fasting. Each stimulus produced the same decrease in blood glucose and weight loss. The polar metabolome was studied using ^1^H NMR, quantifying 59 specific metabolites, and untargeted LC-MS on approximately 5000 features. Results were analyzed by multivariate analyses, using both principal component analysis (PCA) and partial least squares-discriminant analysis (PLS-DA), to identify global metabolic patterns, and by univariate analyses (ANOVA) to assess individual metabolites. The results demonstrated that while there were some similarities in the responses to the two stimuli including decreased glucose, ADP, and glutathione, they elicited distinct metabolic states. The metabolite showing the greatest difference was O-phosphocholine, elevated in ricin-treated animals and known to be affected by the pro-inflammatory cytokine TNF-α. Another difference was the alternative fuel source utilized, with fasting-induced hypoglycemia primarily ketotic, while the response to ricin-induced hypoglycemia involves protein and amino acid catabolism.

## 1. Introduction

Ricin is a type 2 ribosome inactivating protein derived from the castor bean plant, *Ricinus communis*. It assembles as an A-B toxin, in which the B chain, a galactose-specific lectin, binds promiscuously to glycans on the surface of many cells, while the A chain encodes an N-glycosylase that cleaves 28S rRNA at a specific site (adenine 4324) of the large ribosomal subunit. After internalization, ricin is either expelled by membrane blebs, degraded in the phagolysosome, or transported in a retrograde manner from the endosome to the Golgi, endoplasmic reticulum and finally to its site of action in the cytoplasm. The A chain glycosylase is highly active and can cleave up to 1500 ribosomes per minute [1,2,3,4,5,6,7,8]. Ricin toxin is considered a bioterrorism threat because castor beans may be easily obtained, the toxin can be extracted in high purity using common materials, toxic activity is stable for long periods of time at room temperature, and it is highly toxic, LD_50_ < 15 µg/kg by aerosol or injection [9,10,11,12,13,14,15,16]. The ricin A chain is also a potential pharmaceutical, as the toxic moiety of immunotoxins. Our lab has been studying both aspects of ricin, seeking to develop antibodies that can be used to protect against ricin toxicosis [17,18,19] and developing ricin-based anti-HIV immunotoxins [20,21,22]. In our studies of Ab-mediated protection from ricin toxin, we sought to identify a more humane endpoint for studies of protective efficacy than the then standard death-as-endpoint. In seeking biomarkers for ricin toxicosis, we performed a blood chemistry panel at different time points following parenteral administration of ricin. We were surprised to observe that the single abnormal laboratory value was the development of a rapid and profound state of hypoglycemia [23]. We and others have used hypoglycemia as a surrogate marker for ricin toxicosis [17,24,25].

The mechanisms underlying the development of hypoglycemia following systemic administration of ricin have not yet been elucidated. The in vivo effects of ricin administration have been studied for more than a century [26]. The “syndrome” associated with ricin toxicosis is highly dependent upon the route of administration, although inflammation is prominent regardless of the route [15,27,28,29,30]. Localized injection (e.g., intramuscular) results in tissue necrosis as well as systemic effects [26,31,32,33]. Aerosol exposure induces pneumonitis with profound neutrophil influx, marked inflammatory response, fluid accumulation, and respiratory failure [15,16,29]. Oral ingestion of ricin toxin has not been associated with any signs in experimental animals unless animals were gavage fed or fasted, although human ingestion of intact castor beans can produce GI symptoms and even death [10,34,35,36,37,38,39]. To understand the mechanism(s) of hypoglycemia following parenteral administration of ricin to mice, we investigated the function of the major regulators of blood glucose in animals receiving parenteral ricin, published as a companion manuscript to this [40]. We showed that the hypoglycemia likely results from suppression of the transcription of hepatic glucose-6-phosphatase, mediated by TNF-α. Because the final step in gluconeogenesis and glycogenolysis is blocked, the liver is unable to respond to the systemic hypoglycemia by producing glucose de novo or from its glycogen stores.

Hypoglycemia can cause life-threatening consequences and may result from different causes including overtreatment of diabetes mellitus with insulin and other blood glucose lowering agents, inherited deficiencies in glucose metabolism or transport, endocrine disorders, toxic exposures, Gram-negative sepsis, tumors, and prolonged fasting. In this study we have examined the metabolic landscape of the hypoglycemic state induced in the liver by ricin administration. Because the administration of ricin may result in decreased feeding, we used animals that had been fasted as controls. Mice were sacrificed at various times following injection of ricin or initiation of fasting, and polar metabolites were extracted from liver tissue. Specific metabolite profiles were identified and confirmed by ^1^H NMR spectroscopy. LC-MS was used to track global changes in the small molecule profile, as well as to confirm metabolite identification associated with 1D ^1^H NMR spectra. Our analyses showed that whereas fasting and ricin administration both resulted in similar degrees of weight loss and hypoglycemia, the metabolic patterns observed in the livers of these mice were markedly different. Ricin treatment resulted in changes that can be attributed to known effects of TNF-α on cellular enzymes. Amino acids appear to be the main alternative energy source following ricin treatment, whereas in fasted mice there was a switch to ketone body formation. Ketone bodies were observed in response to ricin, but to a much lesser extent. Overall, these results demonstrate that ricin and fasting induce distinctly different metabolic states in the hypoglycemic mice.

## 2. Results

### 2.1. The Effect of Ricin Treatment and Fasting on Weight Loss and Blood Glucose Concentration

One-hundred and seven mice received a lethal (30 µg/kg) or sublethal (0.6 µg/kg) dose of ricin intraperitoneally, were fasted, or served as controls. Mice were sacrificed at the following intervals 2 h (ricin), 8 h (ricin, fasting), or an overnight time course ranging from 18 h to 23 h (control, ricin, fasting). Unexpectedly, four mice in the 23 h high dose ricin group succumbed to the toxin prior to or during the euthanasia process and were excluded from analyses. A second overnight group of 10 mice was injected with 30 µg/kg ricin, and then sacrificed 18 h post ricin exposure. The results from the 16 mice in these two groups were pooled for analyses, and called the “overnight” time point. Using robust Mahalanobis distance measures in the first two components, one mouse in the overnight ricin group was determined to be a biological outlier and removed from the analyses [41]. The removal of this animal did not affect the overall conclusions nor data discussed below.

To compare the effects of fasting and ricin-induced hypoglycemia, blood glucose and weight were measured for each animal prior to and following treatment. Percent change (%Δ) was calculated, and the average change for each group was compared to control to determine significance (Figure 1). Overall, we observed similar decreases in blood sugar following either ricin injection (mean decrease overnight of 55.6% in the high dose group) or fasting (47.8%) Low-dose ricin induced no hypoglycemia at 8 h, and less profound hypoglycemia overnight (39.8%). The overnight weight loss in the fasted group of mice was surprising, averaging almost 12%. Mice are coprophagic, and when dissected the bowel normally contains numerous feces. The feces were absent in the fasted mice, most likely due to the cage change immediately prior to the fast. This difference largely accounted for the weight loss. There was no difference in these responses in male vs. female mice.

### 2.2. Analyses of Hepatic Metabolites in Hypoglycemia Induced by Ricin Administration or by Fasting

To characterize the metabolic effects of fasting and ricin exposure, ^1^H NMR-based metabolomics experiments were performed on polar small molecules extracted from mouse liver tissue, identifying and quantifying 59 metabolites from individual mice in all groups (Appendix A). Figure 2 presents representative ^1^H NMR spectra used to identify and quantify the polar metabolites. The identities of all metabolites were confirmed by spiking samples with purified compounds, 2D NMR analyses, and/or by LC-MS. The metabolite profiles were analyzed using two and three-dimensional PCA and PLS-DA multivariate, and univariate statistical analyses, to assess differences in the ricin-treated and fasted mouse groups at 8 h and overnight post treatment. The 2D-PLS-DA scores plots shown in Figure 3A demonstrate the separate clustering of the different treatment groups at both 8 h and overnight. In discussing the differences between ricin-treated and fasted mice, we will focus on the overnight time point, since these are most pronounced. Figure 3A shows that the ricin-treated group separates from the fasted and control group primarily along the component 1 dimension of the PLS-DA scores plot, while the fasted group is most distinct from control and ricin treated groups along component 2. Principal Components (PC) 1 and 2 account for ~40% of the variance. Metabolites of importance contributing to components 1 and 2 for the overnight treatment groups are shown in Figure 3B. Glucose and ADP dominate both components 1 and 2, with O-phosphocholine contributing to component 2. PCA of the differences between the two overnight groups are shown in Figure 4A, as 3D plots. The results of this unsupervised PCA support observations resulting from PLS-DA, demonstrating that the metabolite profiles of both ricin-treated and fasted mice differ from control and from each other. PCA and PLS-DA comparing results by sex demonstrated no differences.

Analysis of variance (ANOVA) was employed to identify significant (*p* < 0.05) liver metabolites whose concentration changes contributed to the distinct metabolite profiles of the three groups: control, fasted, and ricin-treated (Figure 5, Appendix A). These analyses resulted in the identification of 43 liver metabolites with significant differences in metabolite levels among the three groups. Several patterns of changes were apparent. (1) Thirteen metabolites were found to be increased in both ricin-treated and fasted mice compared to controls, although some to different degrees. These included: small molecule reporters of cellular energetics (ADP, NAD+, NADP+, UMP); ketone body (3-hydroxybutyrate); metabolites involved in nitrogen metabolism including the amino acids aspartate, glutamate, glutamine, ornithine; gut microbiome derived compounds (benzoate, formate); oxypurinol, and choline. (2) Ten metabolites were found to be decreased in both ricin-treated and fasted mice compared to the control group, and included: glucose, ATP, glutathione, betaine, sarcosine (a close relative of glycine, also known as N-methyl glycine, which along with betaine relates to one carbon metabolism), cadaverine (polyamine), fumarate, xanthosine (a product of purine degradation), alanine, and pyruvate (the latter two connected through transamination reactions). (3) The only metabolite increased in fasted and unchanged/or slightly perturbed in ricin was 3-hydroxyisobutyrate (a product of valine degradation and a marker of hepatic mitochondrial fatty acid oxidation). (4) Seven metabolite changes were unique to ricin-treated mice compared to control. Metabolites whose levels were higher in the ricin-treated group included: histidine, lysine, UDP-glucosamine, acetyl-carnitine (the latter being a potential reporter of fatty acid mitochondrial transport for b-oxidation). Those lower in concentration included tyrosine, niacinamide, and inosine. (5) Ten metabolites changed from control in opposite directions in fasted and ricin-treated mice. Those metabolites whose levels were higher in the ricin treated group but lower in the fasted group (compared to the control group) included: O-phosphocholine, UDP-N-acetylglucosamine, valine, urea, and malate. Similarly, metabolites that were lower in the ricin group but higher in the fasted group included cholate, glycerol, glycine, and uridine.

Untargeted HPLC-MS was performed to obtain a larger view of the metabolic changes associated with ricin injection and fasting. The high sensitivity of LCMS facilitates tracking large numbers of small molecules. Therefore, we reasoned that it would be a good technique for corroborating the overarching patterns and shifts in metabolic profiles identified by NMR. MS features were first filtered based on the interquartile range of the intensities of each feature, leaving ~5000 features within the second and third quartiles for analysis. PLS-DA of these data (Figure 4B) confirmed that at both 8 hr and overnight the broad metabolic profiles of ricin-treated and fasted mice are separate from one another, and from control (at 8 h *p* = 0.099, overnight *p* = 0.01). The differences between the groups increased with time.

### 2.3. Temporal Analysis of Metabolic Changes Following Ricin Administration

Metabolite profiles of ricin-treated mice collected 2 h, 8 h and overnight post treatment were analyzed using PLS-DA and ANOVA, to characterize the progression of liver metabolome changes over time compared to the control group. Figure 6 reports the PLS-DA scores plots of this time course analysis. From these data, metabolite profiles of the ricin-treated mice diverge further from control as a function of time, with the overnight ricin treated group being the farthest from the control group and the 2 h ricin treated group being the closest. The largest separation was along PC1, which accounted for 26.7% of the variance, while PC1 and PC2 together accounted for 43.4% of the variance. Variable of Importance (VIP) score plots for components 1 and 2 of the PLS-DA (Figure 6B) identified 15 significant metabolites (VIP > 1.2). Glucose was the greatest contributor to both PC1 and PC2, followed by ADP. Other important metabolites included lactate, O-phosphocholine, creatine, malate, UDP-N-acetylglucosamine, alanine, xanthosine, aspartate, inosine, UMP, O-acetylcarnitine, glutamate, and NADP+.

ANOVA was employed to identify significant (*p* < 0.05) liver metabolites whose level changes contributed most to the characteristic metabolite profiles of the distinct ricin-treated groups over time compared to control (Figure 7). The ANOVA of the time course data demonstrated significant level changes in 54 metabolites (out of 59 identified), with different patterns of changes apparent. Because hypoglycemia was not observed at the first time point (2 h), alterations at this time reflect drivers of the process, whereas changes at the two later points reflect both causes and responses to hypoglycemia. Levels of some metabolites changed at the initial time point and remained similarly altered at all times. These metabolites included the microbial metabolites benzoate and formate (both increased); nucleotides (ATP levels decreased over time while levels of AMP, ADP, UMP and UDP glucose increased); NAD+, NADP+, O-acetylcarnitine (all increased), and inosine (decreased). Levels of other metabolites remained unaltered initially and then rose or fell at the final time point(s). Metabolites whose levels rose at later time points included creatine, urea, O-phosphocholine, and UDP-N-acetylglucosamine, and those whose levels decreased corresponded to glycine and tyrosine. Many of these were the same metabolites that differentiated the hypoglycemia induced by ricin from that observed following fasting, and suggests that the effects of protein catabolism and TNF-α were not manifest until later in the time course. Many metabolites exhibited a steady progression of changes with time. Rising levels were observed for aspartate, choline, histidine, malate, and ornithine. Metabolites whose levels progressively declined with time included alanine, betaine, cadaverine, cholate, glucose, glutathione, glycerol, inosine, lactate, sarcosine, and xanthosine. Others had less consistent patterns with variation in the magnitude of changes over time, including several that were increased at 2 h and 8 h, but fell towards normal at the overnight time point. In this category were glutamate, glutamine, 3-hydroxyisobutyrate, oxypurinol, and phenylalanine.

## 3. Discussion

Hypoglycemia is a clinically important phenomenon, with implications for the management of patients with diabetes mellitus, inborn errors of metabolism, adrenal insufficiency, and endotoxin-induced shock, as well as playing a role in the purported health benefits of intermittent fasting and caloric restriction [42,43,44,45]. In seeking to understand the metabolic states underlying hypoglycemia and unravel causes from homeostatic responses, it is useful to examine hypoglycemia induced by distinct causes. In this manuscript we have compared the effects of fasting with those of systemic ricin administration on the hepatic metabolome of mice. The hypoglycemic effects of fasting are well known [42,45]. We first described ricin-induced hypoglycemia [23], and in a concurrent manuscript, we show that hepatic utilization of stored glucose is impaired by suppression of glucose-6-phosphatase expression, most likely mediated by TNF-α [40]. While both fasting and ricin induced similar degrees of hypoglycemia and weight loss, our metabolomic analyses demonstrated that these stimuli induced changes corresponding to two distinctly different metabolic states, although common effects in response to hypoglycemia were also noted.

To draw the conclusion that ricin and fasting-induced hypoglycemia are distinct metabolic states, we employed widely used dimensionality reducing multivariate statistical analyses (PCA, PLS-DA) to analyze overall metabolic patterns. ANOVA was employed to provide further information and to evaluate concentration change patterns of individual metabolites across treatment groups. PCA is considered an unsupervised multivariate statistical method, meaning the data is projected into two- or three-dimensional plots to identify underlying patterns and whether these patterns result in distinct clustering of the different treatment groups. Further analysis included PLS-DA, which as a supervised multivariate statistical analysis, evaluates the extent of the differences between the groups as a result of their distinct metabolite profiles. As PLS-DA modeling invokes group differences by nature of being a supervised multivariate statistical method, validation steps are necessary to confirm the PLS-DA modeling to ensure that the group separations shown in the PLS-DA scores plots are real and not due to an overfit of the data. Validation metrics include coefficients of discrimination (R^2^ and Q^2^), permutation tests, classification error rate (CER) and area under the receiver operating characteristic (AUROC), which are reported in the figure legends associated with the PLS-DA scores plots (Figure 3 and Figure 6). We performed both PLS-DA and PCA at two time points. Both analyses demonstrated the differences in the metabolic states (Figure 3 and Figure 4). PLS-DA models which have passed the validation metrics yielded information about metabolites that are the most important discriminators of the different treatment groups, as seen by those metabolites with VIP scores > 1.2 (Figure 3 and Figure 6). ANOVA resulted in the identification of 54 metabolites (out of 59 metabolites identified by NMR) with levels that were significantly different (*p* < 0.05) in the different treatment groups. These data not only demonstrated that ricin treatment results in a different liver metabolic state than that induced by fasting, but also highlighted differences in the range of metabolite concentrations observed in these different metabolic states (Appendix A, Figure 5 and Figure 7). To further validate our conclusion that the two stimuli result in distinct metabolic states, we extended our analyses beyond the 59 metabolites identified by ^1^H NMR to MS “features”, i.e., LC-MS peaks at specific *m/z* ratios, that number in the thousands, and again demonstrated that ricin treatment and fasting result in distinct metabolic states (Figure 4B).

In evaluating differences observed after fasting or ricin treatment, several trends emerged (Figure 3B and Figure 5). Despite comparable blood glucose levels, hepatic glucose levels were much lower in the ricin-treated group compared to the fasting group, which would indicate a more pronounced decline in glucose stores following ricin administration, and likely contributing to the fatal outcome following ricin administration. The metabolite demonstrating the greatest difference in concentration between ricin-treated and fasted mice was O-phosphocholine, with an 18-fold higher concentration in the livers of ricin-treated mice. In the Kennedy (CDP-choline) pathway [46,47], O-phosphocholine is the metabolite upstream of the enzyme choline-phosphate cytidylyltransferase, known to be downregulated by TNF-α, and TNF-α has been shown to increase cellular levels of O-phosphocholine in a dose dependent fashion [48,49]. There is evidence that O-phosphocholine offers protection against TNF-induced apoptosis [50]. Another metabolite uniquely altered in ricin-treated mice was UDP-N-acetylglucosamine, the substrate for O-linked protein modifications and an important molecule in cell signaling pathways, including NFκ-B, the mediator of TNF-α-induced effects [51,52,53]. It has been proposed that the ratio of O-phosphocholine to UDP-N-acetylglucosamine serves as a biomarker reflecting glycemic states [54]. Our PLS-DA analyses showed that both metabolites contributed significantly to the progression of metabolic changes observed with time following ricin treatment (Figure 6).

Another important difference between fasting and ricin-induced hypoglycemia appears to be the alternative fuel source utilized in each case. Fasting resulted in significantly higher levels of 3-hydroxybutyrate (ketone), indicating the breakdown of lipids resulting in ketone body formation by the liver, a well-known response to caloric deprivation. In contrast the response to ricin-induced hypoglycemia appears to utilize mobilization and catabolism of amino acid pools to serve as energy source, as evidenced by elevated liver urea and creatine levels in the ricin-treated animals. All together these patterns suggest that the response to fasting-induced hypoglycemia is primarily ketotic in origin, while the response to ricin-induced hypoglycemia involves protein and amino acid catabolism rather than ketone body production (although the latter does take place following ricin treatment but to a much lesser extent).

Glutathione levels were decreased following both fasting and ricin-administration, suggesting that both interventions cause oxidative stress in hepatocytes. Glycine, which is used in the synthesis of glutathione, was higher in fasted compared to control, but lower in ricin compared to control, which could suggest a greater oxidative stress in the ricin-treated group. This interpretation is consistent with higher levels of NAD+ and NADP+, and lower levels of niacinamide in ricin-treated animals compared to fasted. Lower levels of niacinamide would indicate that the ricin treated group used this precursor to increase synthesis of NAD+/NADP+. Glycine is involved in other cellular processes that may account for the disparate changes following fasting and ricin administration, including folate, one carbon, and nucleotide metabolism. Glycine has been shown to inhibit the effects of TNF-α [55,56]. The decreased levels of glycine following ricin treatment may exacerbate TNF-α’s deleterious effects observed in this condition, further contributing to hypoglycemia.

Despite our highlighting metabolic differences between the two hypoglycemic states, we also observed similarity in the hepatic responses to fasting and ricin treatment. Significant changes in the same direction from control were observed in 23 metabolites following both ricin and fasting, although there were differences in the degree of change. These commonalities may reflect consequences of the hypoglycemic condition or homeostatic mechanisms in response to this life-threatening condition.

We first described hypoglycemia associated with parenteral administration of ricin toxin in the course of biodefense studies [23]. We and others have used hypoglycemia as a surrogate marker for ricin toxicosis [17,24,25]. Hypoglycemia has also been observed, but not commented upon, in porcine [32] and human [57] intoxications. Metabolomics investigations of ricin intoxication have been undertaken by others, but used different species, methods of exposure, analyzed different tissues, employed different detection methodology, and identified different metabolites [58], making direct comparisons of results difficult.

A thorough understanding of metabolome changes resulting from induced hypoglycemia in experimental animals can be translated to the development of therapies that protect against damage caused by hypoglycemia, or mimic the beneficial effects of intermittent fasting and caloric restriction. In the studies reported here, we have begun this process. A number of experimental steps need to be completed before the analysis may be considered thorough. The first would be to include insulin-induced hypoglycemia to the stimuli under study. We would also extend the analyses to additional tissues, including pancreatic islet cells, adipose tissue, cardiac muscle and brain. Future studies should also include analyses of lipids and other non-polar molecules using both NMR and mass spectrometry to identify additional metabolites. Coupling analyses of gene expression (e.g., RNAseq) with metabolite data would allow detailed pathway analyses and provide greater insight into the metabolic causes and consequences of hypoglycemia.

## 4. Materials and Methods

### 4.1. Animal Handling

BALB/c mice were bred (using animals originating from Jackson Labs, Bar Harbor, ME, USA) and housed at the Montana State University Animal Resources Center (Bozeman, MT, USA). 52 male and 55 female mice were studied (Appendix A). Inbred mice limit variability and the strain specific inflammatory response is well documented for BALB/c mice [59]. Animals were fed irradiated rodent chow (LabDiet 5053; Fort Worth, TX, USA). Male and female mice were housed separately with five mice per cage and were studied at 10 to 11 weeks of age. Each treatment group contained approximately equal numbers of males and females. All animal care and experimentation were approved by the MSU Institutional Animal Care and Use Committee. Immediately prior to treatment, baseline weight and blood glucose were measured. To measure blood glucose, mice were bled from the saphenous vein and blood was transferred to a test strip. Blood glucose was measured using a handheld CVS Health Advanced glucose meter and test strips (Agametrix; Salem, NH, USA) following the manufacturers protocol. Mice were then injected via intraperitoneal route with either ricin (Vector Labs, Burlingame, CA, USA, at 0.6 or 30 µg/kg) or saline in 10 µL per g body weight. Mice in all but the fasted group were given food and water ad libitum. Fasted mice were transferred to new cages with fresh bedding where food was withheld while giving water ad libitum. Immediately prior to sacrifice, mice were weighed, and blood glucose measured. Mice were sacrificed at 2 h (ricin), 8 h (ricin and fasted), and overnight (16–22 h, control, ricin, fasted). If any animals were found to be exhibiting signs of pain or suffering, as judged by animal technicians before those times, they were euthanized. Prior to euthanasia, animals were anaesthetized with isoflurane, bled by cardiac puncture and plasma stored, and sacrificed by severing of the cervical spinal cord. Liver and muscle were immediately excised and stored at −80 °C.

### 4.2. Metabolite Extraction for NMR Analysis

Polar metabolites were extracted from whole livers using chloroform-methanol-water [60]. Briefly, frozen livers were homogenized in liquid nitrogen using a mortar and pestle. Liver powder was weighed (60–80 mg) into 2 mL microcentrifuge tubes. All samples were kept on ice until use. To each microfuge tube 900 µL of ice-cold 1:2 methanol/CHCl_3_ was added and the tissue was mixed into suspension, then lysed using a FastPrep-24^TM^ 5G homogenizer (MP Biomedicals; Solon, OH, USA) using silica beads and a lysis setting of 6.0 m/s for 2 × 40 s. Tubes were placed back on ice and 200 µL of ice-cold H_2_O added, and the homogenization step was repeated. Samples were then placed at −20 °C for approximately 45 min to precipitate protein, centrifuged at 14,000× *g* for 10 min (8–10 samples at a time to prevent remixing of phases). 250 µL of upper polar phase were transferred to a 1.5 mL tube and dried overnight using a Savant™ Universal SpeedVac™ vacuum system (ThermoFisher; Waltham, MA, USA) with no heat.

### 4.3. H-NMR Analyses

Dried metabolite mixtures were re-suspended in 600 µL of NMR buffer containing 0.25 mM 4,4-dimethyl-4-silapentane-1-sulfonic acid (DSS), 0.4 mM imidazole, 90% H_2_O/10% D_2_O, 25 mM sodium phosphate, pH 7. Once redissolved, the metabolite mixtures were transferred into 5 mm NMR tubes (Bruker; Billerica, MA, USA). All one dimensional (1D) ^1^H NMR spectra were recorded at 298 K using a Bruker AVANCE III solution NMR spectrometer operating at 600.13 MHz (^1^H Larmor frequency) and equipped with a 5 mm liquid-helium-cooled TCI cryoprobe with Z-gradient and a SampleJet^TM^ automatic sample loading system (Bruker). 1D ^1^H NMR data were acquired using the Bruker supplied 1D excitation sculpting water suppression pulse sequence ‘*zgesgp*’ with 256 scans, a ^1^H spectral window of 7200 Hz, 64 K data points, a dwell time interval of 69 μs between points, and D1 delay of 2 s, amounting to a total relaxation recovery delay between scans of ~5.3 s. The data were first processed with the Bruker TOPSPIN version 3.5 software using the first DSS peak for chemical shift reference, a line-broadening function of 0.3 Hz, and application of a polynomial function (qfil, 0.2 ppm width) to remove the residual water signal. Spectral phasing and baseline correction were done using the Chenomx NMR software suite (version 8.3). Identification and quantification of metabolites were carried out using the Chenomx profiling module and its associated 600 MHz small molecule reference spectral database, as well as the Human Metabolome Database (HMDB). The DSS (0.25 mM) peak was used as an internal standard for metabolite quantification, while imidazole NMR signals were used to correct for small chemical shift changes arising from pH variations between samples. Tables of metabolite concentrations (mM) generated by profiling in Chenomx were exported to a .csv file and normalized to dried liver weights, resulting in final concentration tables in picomoles per mg dried liver tissue (Appendix A). Validation of select metabolite IDs that were annotated using Chenomx was accomplished using 2D ^1^H-^1^H (TOCSY) NMR, by spiking the samples with pure metabolite standards (when available), and/or by mass spectrometry. 2D ^1^H-^1^H TOCSY spectra were acquired for representative samples using the Bruker-supplied ‘mlevphpr.2/mlevgpph19′ pulse sequences and the following experimental parameters: 256 t1 points; 2048 t2 data points, 2 s relaxation delay, 32 scans per t1 interval, ^1^H spectral window of 6602.11 Hz, and 80 ms TOCSY spin lock mixing period. 2D ^1^H-^1^H TOCSY spectra were processed and analyzed using Topspin software (Bruker version 3.2).

### 4.4. Statistical Analysis of the Metabolite NMR Data

The NMR-based metabolite data were uploaded to the MetaboAnalyst v4.0 web server and used for multivariate statistical analysis. Metabolite concentrations were log-transformed and auto-scaled (mean centered divided by the standard deviation of each variable) prior to univariate and multivariate statistical analysis. Student *t*-test, hierarchical cluster analysis, PCA, and PLS-DA were performed to identify potentially distinct liver metabolite patterns that separate the different mouse treatment groups. VIP plots were employed to assess the importance of each metabolite in the PLS-DA models. Validation and robustness of group separations in the PLS-DA scores plots were assessed using Q^2^ and R^2^ statistics, and the (B/W) permutation test function of MetaboAnalyst, which, using 2000 permutation steps provided a measure of the PLS-DA group separation significance. Additionally, PLS-DA classification separation was assessed using CER and AUROC analysis in the ‘MixOmics’ package in R (Bioconductor). One-way ANOVA applying Tukey’s post hoc analysis was performed on all metabolites determined to be statistically significant (*p* < 0.05) based on concentration differences. To evaluate if multivariate outliers were present, which would lead to subsequent violations of multivariate normality, an analysis of robust Mahalanobis distances of the first two components was performed using the ‘MVN’ package in R [41,61]. One sample was found to be an extreme observation, this sample was removed from subsequent analyses.

### 4.5. Metabolite Extraction for LC-MS Analysis

30 mg of frozen liver powder (prepared as above) were transferred into 2 mL microcentrifuge tubes. Tubes were kept on ice unless otherwise specified. To each tube, 500 µL of ice-cold acetone was added (Fisher Scientific, Hampton, NH, USA), as well as three Zirconia beads (1.0 mm diameter, 5.5 g/cc density) (Biospec Products; Bartlesville, OK, USA) for liver tissue lysis. Tubes were placed in the FastPrep-24^TM^ 5G instrument (MP Biomedical, Irvine, CA, USA) and tissue was lysed and homogenized as described above. Tubes were then placed in a sonicator bath (42 kHz) (Fisher Scientific; Pittsburgh, PA, USA) for 15 min, and then centrifuged at 14,000× *g* for 10 min, resulting in the formation of a protein precipitate at the bottom of the tube. The supernatant was transferred to a 2 mL Eppendorf tube. To the precipitated pellet, 500 µL of ice-cold 5:1 methanol/H_2_O solution was added. Tissue lysis was repeated with the FastPrep-24 bead beater, followed by another round of sonication and centrifugation. The supernatant was transferred to the sample tube containing the nonpolar acetone fraction and dried in the SpeedVac™ overnight. Dried metabolite pellets were resuspended in 1:1 acetonitrile/H_2_O with 1% formic acid and vortexed for 30 s, or until the entire pellet was suspended in solution. Samples were sonicated once more for 15 min and centrifuged at 14,000× *g* for 10 min. 180 µL of eluate was then transferred to vials for LC-MS analyses.

### 4.6. LC-MS Analyses

The metabolite extracts were run on an Agilent 1290 Ultra High-Performance Liquid Chromatograph (UHPLC)-coupled to an Agilent 6538 Quadrupole-Time of Flight (Q-TOF) mass analyzer (Agilent; Santa Clara, CA, USA). An autosampler was used to inject 5 µL of sample onto a BEH-HILIC column (Waters Inc., Millford, MA, USA) with a pressure limit of 600 bar. The mobile phase consisted of A (HPLC grade water with 0.1% formic acid) and B (Acetonitrile with 0.1% formic acid) set to a flow rate of 0.400 mL/min. Gradient elution proceeded as follows: 100% B (initial), 55% B (15 to 18 min), 30% B (18 to 21 min), 100% B (21 to 25 min). The column temperature was held constant at 40 °C. Mass spectra were acquired in positive ionization mode.

### 4.7. Statistical Analysis of Mass Spectrometry Data

Raw LC-MS data were converted to mzML format using the ProteoWizard msconvert program (GitHub, Palo Alto, CA, USA). Mass spectral feature identification was performed with XCMS Online v3.7.1, and resulting data was uploaded to MetaboAnalyst v4.0 for multivariate analysis. Mass spectral features were sorted and selected based on interquartile range [62]. This selected the central ~5000 features for statistical analysis, with data auto-scaled, log-transformed, and normalized by sum, followed by PCA and PLS-DA. PLS-DA models were validated using the cross-validation statistics Q^2.^ and R^2^, and model significance was assessed by permutation test as described above.

## Figures and Tables

**Figure 1 toxins-14-00815-f001:**
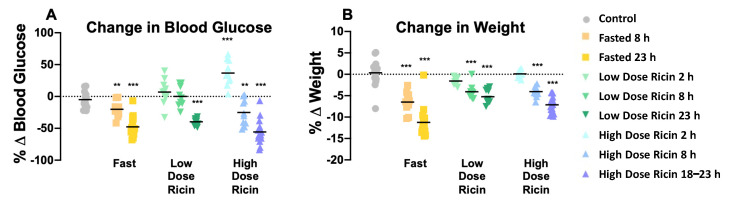
**Blood glucose and weight change resulting from treatment.** Baseline blood glucose and weight for each mouse were measured before treatment, and again after, preceding sacrifice. %Δ = ((post-treatment weight − initial weight) / initial weight) × 100. Symbols represent individual mice, with the mean shown as horizontal black bars. (**A**) Percent change in blood glucose over the period of treatment. (**B**) Percent change in weight that occurred in each treatment group. Asterisks indicate that changes were significantly different compared to control. Significance was determined by unpaired *t*-tests, ** indicates a *p*-value ≤ 0.01, and *** indicates *p*-value ≤ 0.001.

**Figure 2 toxins-14-00815-f002:**
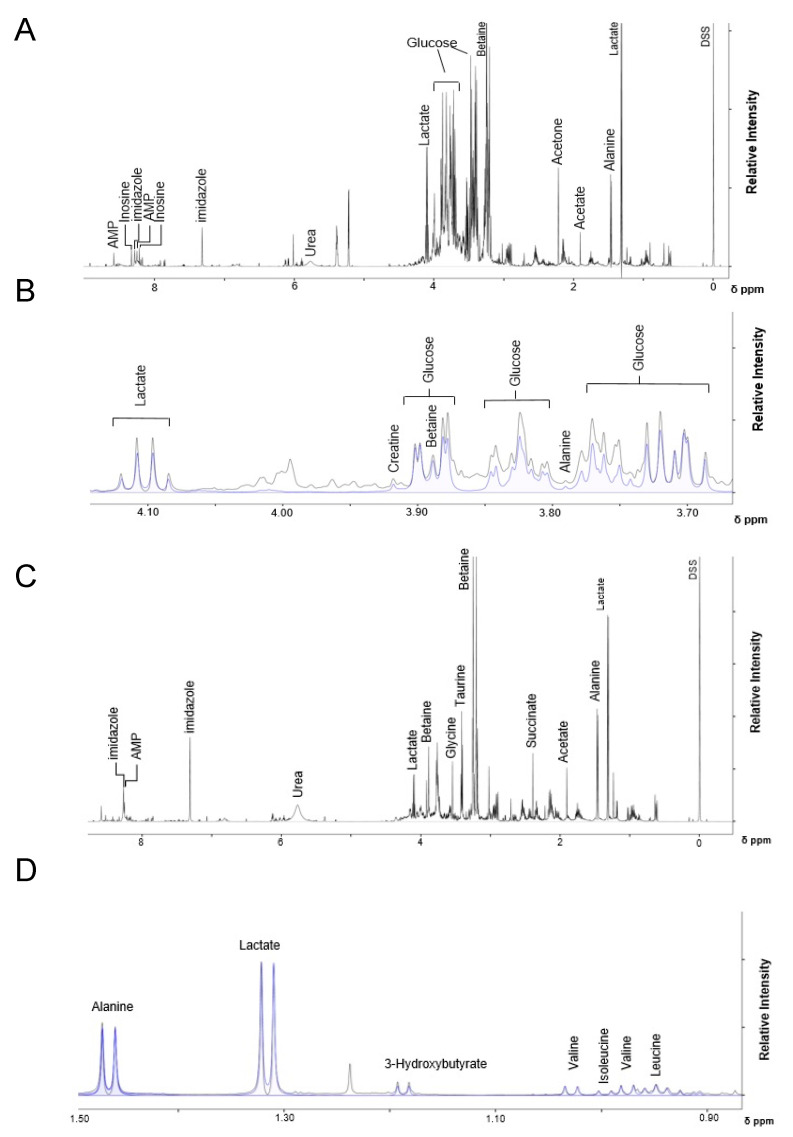
**Representative 1D-**^1^**H-NMR spectra of polar metabolites from control and ricin-treated mice livers.** Spectra were recorded on a Bruker Avance III 600 MHz spectrometer. Chemical shift in ppm is denoted on the x-axis, and relative signal intensity is denoted on the y-axis. The 1D ^1^H-NMR spectra is shown in black, and the spectral peaks for labeled metabolites are shown in blue. (**A**) 0.0–8.0 ppm region of the 1D-^1^H-NMR spectra obtained from polar molecules extracted from control mice, and (**B**) expanded view of the 3.70–4.10 ppm region of the same spectrum in A. (**C**) 0.0–8.0 ppm region of the 1D-^1^H-NMR spectra from polar molecules extracted from overnight ricin-treated mice, and (**D**) expanded view of the 0.90–1.50 ppm region of the same spectrum in (**C**).

**Figure 3 toxins-14-00815-f003:**
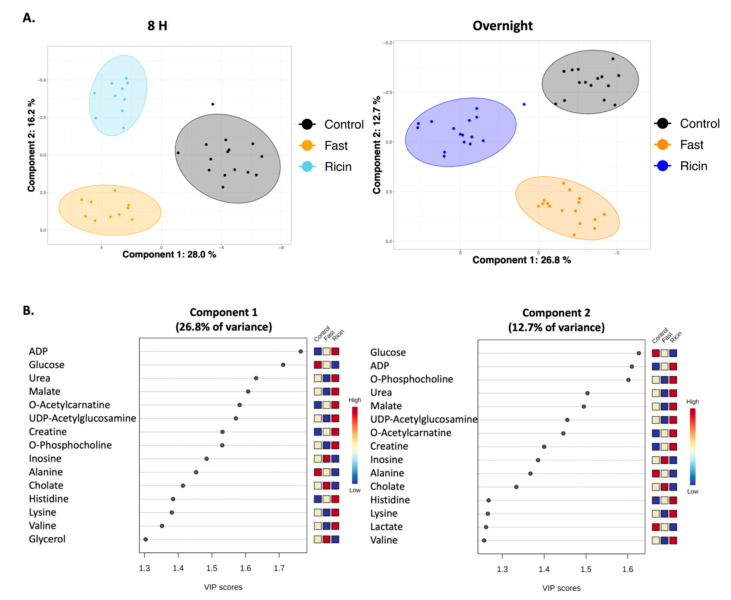
**Two dimensional PLS-DA showing differences among control, fasted, and ricin-treated groups**. (**A**) Plots showing differences among the groups at 8 h and 18–23 h. Filled colored circles represent samples from individual mice, the shaded areas are 95% confidence limits. Validation statistics are as follows: 8 h graph: Q^2^ = 0.87 (component 2), R^2^ = 0.90 (component 2), permutation *p* < 5e−0 (*n* = 2000), and CER: 0.0 (component 2). Overnight graph: Q^2^ = 0.85 (component 2), R^2^ = 0.90 (component 2), permutation *p* < 5e−0 (*n* = 2000), and CER: 0.017 (component 2). (**B**) **Metabolites of importance in components 1 and 2 of PLS-DA of overnight fasted vs. ricin treated mice**. The metabolites contributing most to each component of the PLS-DA shown in A are ranked according to importance. A score >1.2 is considered significant. Relative metabolite levels for each group are indicated to the right of the VIP graph.

**Figure 4 toxins-14-00815-f004:**
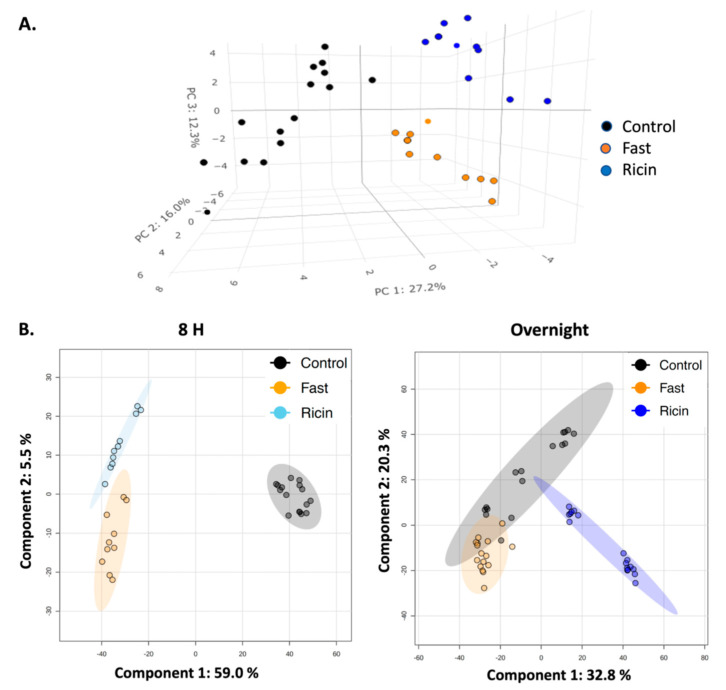
**Confirmation of differences between fasted and ricin-treated mice.** (**A**) **3D PCA plot of metabolites detected by NMR in overnight fasted vs. ricin-treated mice**. The unsupervised PCA plot confirms the PLS-DA plots showing that both ricin treatment and fasting induce metabolic changes that are distinct from untreated control mice, and from each other. (**B**) **PLS-DA analysis of MS features**. Comparisons of fasted, control, and ricin-treated mice at 8 h and overnight post-treatment were performed utilizing approximately 5000 MS features falling between the first and third quartiles. Validation criteria are: for the 8 h time point Q^2^ = 0.98 (Component 2), R^2^ = 0.99 (Component 2), and Accuracy = 0.91 (Component 2); at the overnight time point Q^2^ = 0.88 (Component 4), R^2^ = 0.93 (Component 4), and Accuracy = 1.0 (Component 4).

**Figure 5 toxins-14-00815-f005:**
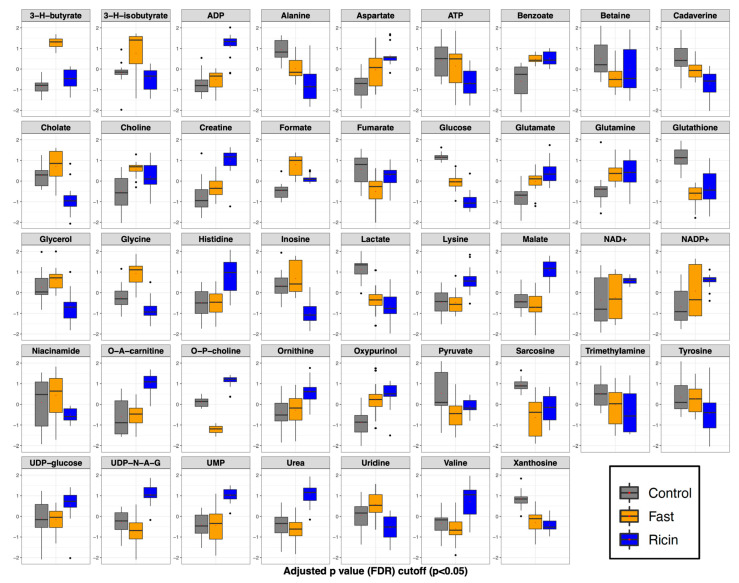
**ANOVA demonstrating significant metabolite changes among control, fasted, and ricin-treated mice.** Box and whisker plots for metabolites with statistically significant differences by one-way ANOVA, with Tukey’s post hoc analysis. Metabolites are listed alphabetically. The vertical axis is the metabolite concentration, log transformed and autoscaled (mean-centered and divided by the standard deviation of each variable). The horizontal bar within each box represents the sample median, the box represents the range from quartiles 1 to 3. The whiskers represent 1.5 times the IQR (interquartile range). The black dots represent outliers between 1.5 and 3 X IQR.

**Figure 6 toxins-14-00815-f006:**
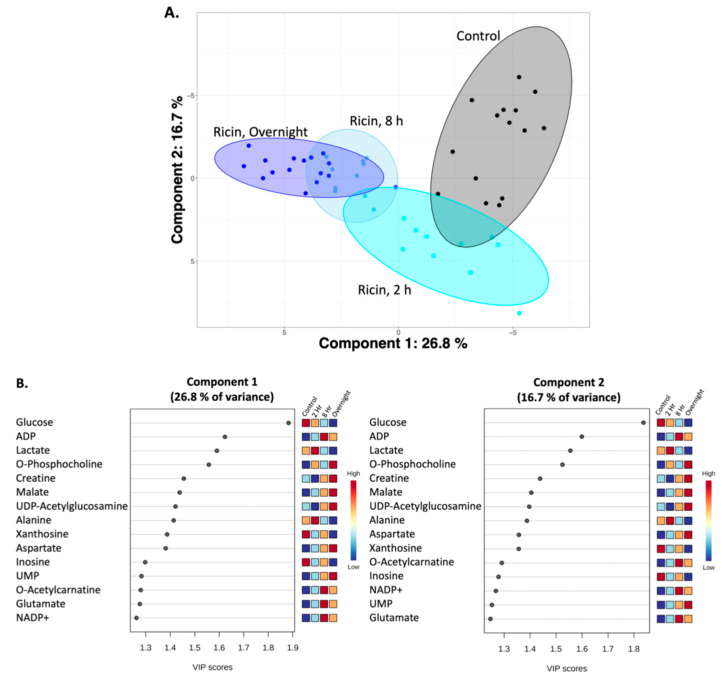
**PLS-DA plot showing progression of metabolic changes with time in ricin-treated mice relative to the control group, as assessed by**^1^**H-NMR metabolomics**. (**A**) **PLS-DA plots**. Validation metrics of the PLS-DA model included satisfactory R^2^ and Q^2^ (0.8) values (Component 5), a significant permutation test *p* value < 0.001 (*n* = 1000), a classification error rate (CER) value of 0.05 (Component 5) and acceptable ROC curve parameter. (**B**) **Metabolites which drive the separation of components 1 and 2**. The metabolites contributing most to the separation along each component of the PLS-DA shown in A are ranked according to importance. A score >1.2 is considered significant. Relative metabolite levels for each time point are indicated to the right of each VIP graph.

**Figure 7 toxins-14-00815-f007:**
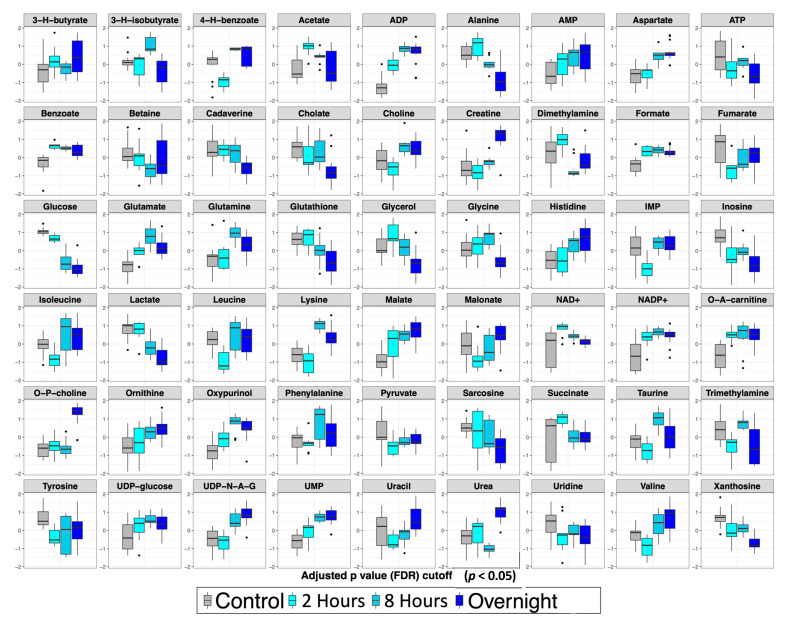
**ANOVA showing significant metabolite changes over time in ricin-treated mice**. Box and whisker plots are as described in the legend to Figure 5.

## Data Availability

These data have been deposited in the Metabolomics Workbench data repository as studies ST001614 (NMR data) and ST001617 (MS data).

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
