# Peer review of "Distinct Metabolic States Are Observed in Hypoglycemia Induced in Mice by Ricin Toxin or by Fasting"

_toxins, 2022, doi:10.3390/toxins14120815_

Round 1

Reviewer 1 Report

In this manuscript the authors study the hypoglycemic state by comparing the effects of fasting with those of systemic administration of ricin on the hepatic metabolome of mice and find that the treatments elicit distinct metabolic states with differences in the alternative fuel source used, with fasting-induced hypoglycemia primarily ketosis, whereas the response to ricin-induced hypoglycemia involves protein and amino acid catabolism.

This is a very thorough metabolome study, and the analysis and discussion of the results are adequate and comprehensive. The images and the results in general, are very well presented and organized.

It is well written and an interesting manuscript in general.

Minor comments

Line 24 Key contribution of this manuscript. Allusions to the companion manuscript should be deleted.

Line 32 and 36 Change “N-glycosidase” by “N-glycosylase”

Line 160 Correct. According to Figure 5, niacinamide is not a metabolite whose level was higher in the ricin-treated group.

Line 204 According to Figure 7, change “UMP glucose” by “UDP glucose”

Line 261 Change “(Table 1, Figures 5 and 7)” by “(Table S1, Figures 5 and 7)”

Reference 40 - The manuscript is only submitted. This must be solved in the final version of this manuscript because reference 40 is cited in text (lines 62 and 231) but the paper cannot be accessed.

Author Response

See attached file for responses to criticisms.

Reviewer 2 Report

The authors have previously identified hypoglycemia as a marker of ricin toxicosis in order to avoid using death as a time point in treatment protection experiments. Here, the authors analyze the metabolic changes in the liver of mice linked to ricin toxicosis-induced hypoglycemia versus fasting hypoglycemia. They conclude that ricin-induced hypoglycemia and accompanying metabolic changes resemble those of TNFalpha effects, a mediator of inflammation during ricin intoxication. As a result of toxicosis, amino-acids appear to be the main energy source whereas fasting leads to using lipids as energy source.

The experiments seem carefully done, metabolisms in tested and control conditions are thoroughly analyzed. Altogether, the results bring a significant contribution to the understanding of metabolic processes during ricin intoxication.

Minor: Please respect space between numerals and time units (6 h instead 6h).

Author Response

(The authors gave the same response as above.)
